# The Assessment of an Effect of Natural Origin Products on the Initial Growth and Development of Maize under Drought Stress and the Occurrence of Selected Pathogens

Joanna Horoszkiewicz [1,*], Ewa Jajor [1], Jakub Danielewicz [1], Marek Korbas [1], Lech Schimmelpfennig [2], Marzena Mikos-Szymańska [2,*], Marta Klimczyk [2] and Jan Bocianowski [3]

1   Institute of Plant Protection—National Research Institute, Węgorka 20, 60-318 Poznań, Poland
2   Grupa Azoty Zakłady Azotowe "Puławy" S.A., Al. Tysiąclecia Państwa Polskiego 13, 24-110 Puławy, Poland
3   Department of Mathematical and Statistical Methods, Poznań University of Life Sciences, ul. Wojska Polskiego 28, 60-637 Poznań, Poland
*   Correspondence: j.horoszkiewicz@iorpib.poznan.pl (J.H.); marzena.mikos-szymanska@grupaazoty.com (M.M.-S.)

**Abstract:** Poland, like other countries in the world, increasingly experiences the ongoing climate change that is a critical yield-limiting factor. The use of biostimulants in agriculture has shown tremendous potential in combating climate change-induced stresses such as drought, temperature stress, etc. They could be a promising tool in the current crop production scenario. Biostimulants are organic compounds, microbes, or amalgamation of both that could regulate plant growth behavior through molecular alteration and physiological, biochemical, and anatomical modulations. They can promote plant growth under various environmental stresses because they have a positive effect, in particular, on plant growth and resistance. There are many products of this type available on the market, including those of natural origin, which are part of the Integrated Pest Management. The ecotoxicity of chemical plant protection products, the negative effects of their use, and the change in regulations make it recommended to use low-risk chemicals and non-chemical methods, that involve the least risk to health and the environment, and at the same time ensure effective and efficient protection of crops. Natural origin biocomponents obtained by the supercritical $CO_2$ extraction of plant material or by fermentation process in bioreactors were tested. Common maize (*Zea mays* L.) was selected as a test plant for growth tests at climate chambers. Results showed that the only supernatant (fermentation broth) obtained with the *Paenibacillus* bacteria (S2) had a positive effect on the germination index (GI > 100%) of maize seeds, compared to the obtained plant seed extracts from the crop of the legume family (*Fabaceae*) (E3) and from the crop of the smartweed family (*Polygonaceae*) (E9) (GI < 100%). The extracts E3, S1 (supernatant obtained with the use of bacteria from the genus *Enterobacter*) and S2 used as a single product and in combination with UAN+S, under optimal conditions of the experiment, had a positive effect on the maize root weight compared to the untreated, while under drought stress, a decrease in the root weight was observed. Moreover, on the basis of the conducted research, differences in the mycelial growth of selected fungi were found. The applied biocomponent S2 of microbial origin extract (supernatant 2) showed a mycelial growth-limiting effect on all tested *Fusarium* fungi isolated from the corn cobs.

**Keywords:** germination; root parameters; antifungal activity

## 1. Introduction

Nowadays, the agricultural sector is facing challenges of rising the productivity to feed the growing global population and increasing the resources use efficiency, while reducing the environmental impact on the ecosystems and human health. Fertilizers and pesticides play a crucial role in agriculture in representing a powerful tool for growers to increase yield and guarantee continuous productivity. In the last years, several technological innovations have been proposed to enhance the sustainability of agricultural production systems, through a significant reduction of synthetic agrochemicals like pesticides and fertilizers. A promising

and environmental-friendly innovation would be the use of natural plant biostimulants (PBs) that enhance flowering, plant growth, fruit set, crop productivity, and nutrient use efficiency (NUE), and are also able to improve the tolerance against a wide range of abiotic stressors [1].

Maize (*Zea mays* L.) is one of the most important cereal crops worldwide. The total harvested area of maize (corn) in Poland in 2020 was 946,060 hectares, with a total production of about 6,694,650 tonnes of grains, and an average yield of 7.08 tonnes/ha [2]. It is important to find alternatives to increase the production of maize per unit of land area. One of the new methods that should be taken into consideration is the use of growth promoters that do not affect humans or the environment adversely.

Drought stress can increasingly diminish yields of important cereals by over 10%; it is still the main limiting factor of food production in numerous countries, affecting several crop plants, such as maize. Lack of water negatively impacts plant growth and development by inducing an array of changes at molecular and cellular levels, translated into alterations in plant physiology and morphology [3].

Using biostimulants to promote plant growth has recently acquired expanding attention worldwide [4]. Biostimulants are able to stimulate nutrient uptake, use efficiency by plants, and improve crop quality [5]. They can increase the activity of rhizosphere microbes and soil enzymes, the production of hormones and/or growth regulators in soil and plants, and the photosynthetic process [6,7]. The addition of biostimulants to plants also modifies the morphology of plant roots in a similar way to indole acetic acid (IAA), suggesting that they induce a "nutrient addition response" that favors the uptake of nutrients via an increase in the absorptive surface area [8]. Biostimulants are used as a strategy to minimize the effects of climatic adversities by allowing the seedlings to express more strongly their metabolic capacity and to have a greater root system development. Biostimulants offer a promising avenue for improving plant growth and productivity, while also reducing the need for synthetic fertilizers and pesticides. However, it is important to note that biostimulants are not a silver bullet solution to plant growth and development, and their effectiveness can vary depending on the plant species, soil type, and environmental conditions. Root systems determine plant water and nutrient uptake and affect plant growth and yield [9]. These effects on growth appear to be featured from the nutritional effect of an additional nitrogen source [10,11]. The mode of action of biostimulants is often unknown and hard to identify, because they derive mainly from complex sources containing several bioactive components that, together, may contribute to specific effects in plants [12–14].

Mycotoxin-producing fungi may play an important role in maize cultivation. During the vegetation season, particularly dangerous for food safety, is the occurrence of fungi from the genus *Fusarium*, such as: *Fusarium fujikuori*, *F. graminearum*, *F. culmorum*, *F. avenaceum*, and others. These fungi can contaminate the raw corn material with secondary metabolites. Common mycotoxins produced by the mentioned-above fungi are: deoxynivalenol, zearalenone, fumonisin, and the T-2 toxin [15,16]. These mycotoxins can negatively affect human and animal health [17–19]. *Fusarium* fungi can occur throughout the vegetation season of maize and can cause seedling blight, fusarium wilts, and fusarium stalk rot. In addition to many registered seed dressings and spraying treatments used during vegetation season, a biofungicide, based on *Trichoderma asperellum*, is registered in Poland to reduce diseases caused by fungi of the genus *Fusarium*. Many studies are being conducted using alternative compounds to limit the growth of the mentioned-above fungi; the following are used for this purpose: extracts, plant origin extracts, and vegetable oils [20]. In this study, we use plant origin extracts and microorganism metabolite extracts. The European Green Deal, approved in 2020 by the European Commission, aims to limit the use of synthetic plant protection products; need for the several modern biofungicides will increase.

The novelty of work is the development of new specialty products used for the benefit of crop production that can be used with liquid nitrogen fertilizers enriched with sulphur. The products contain organic substances of plant or microbial origin that, when applied to plants or the rhizosphere, stimulates the natural processes to enhance or benefit nutrient uptake, nutrient efficiency, tolerance to abiotic stress, or crop quality and yield.

The aim of this research was to evaluate the effect of new specialty products of natural origin on the initial growth and development of maize under the influence of drought stress and the occurrence of selected pathogens.

## 2. Materials and Methods

### 2.1. Laboratory Trial

2.1.1. Seed Germination

Seed germination and root length of maize seedlings were investigated in the laboratory using a completely randomized design. Seeds were germinated at $21 \pm 1$ °C in a dark growth chamber in Petri dishes with a moist blotting paper. Two plant extracts and two microbial supernatants (Table 1) in five different concentrations of 0.25, 0.5, 1, 2, and 4% (*w/v*) were used for tests in an amount of 4 mL per Petri dish. The untreated (control) was distilled water. Seed germinability and root length were determined using four groups of 10 seeds for each treatment as four replicates. Seeds were considered as germinated when the coleoptile and radicle length had grown to about 2 mm. Germinated seeds were counted, and the root lengths were measured after four days. To combine these endpoints (seed germination and root elongation), results were expressed as a Germination Index, in percent of the control (%GI), according to the equation:

$$\%Gl = 100 \times (Gs \times Ls)/(Gc \times Lc) \tag{1}$$

where Gs and Gc are the number of germinated seeds in the sample and control, respectively; Ls and Lc are the lengths (mm) of the roots in the sample and control [21].

**Table 1.** Characteristics of the preparations used in the experiments.

| Symbol | Name | Characteristic |
|:---:|:---:|:---:|
| C | untreated (control) | - |
| E3 | the extract of plant seeds 1 | the extract of a plant seeds from the crop of legume family (*Fabaceae*), method of production: supercritical $CO_2$ extraction |
| E9 | the extract of plant seeds 2 | the extract of a plant seeds from the crop of smartweed family (*Polygonaceae*), method of production: supercritical $CO_2$ extraction |
| S1 | the microbial origin biocomponent 1 | the fermentation broth (supernatant 1) obtained with the use of bacteria of the genus *Enterobacter* |
| S2 | the microbial origin biocomponent 2 | the fermentation broth (supernatant 2) obtained with the use of bacteria of the genus *Paenibacillus* |

2.1.2. Pot Experiments

Pot experiments were conducted in climatic chambers at two conditions (C): the optimal growth condition (photoperiod 16/8 h, temperature 21/16 °C, relative humidity (RH) 60%), and the drought stress condition (photoperiod 16 h/8 h, temperature 30/21 °C, RH < 40%). Water stress was applied after 14 days of maize growing at optimal conditions, and lasted for 7 days. The maize plants under drought stress were not watered during the last 7 days of the pot experiment. The experiment was stopped when the maize leaves started to wither. The soil for treatments was prepared by mixing deacidified peat (pH = 5.5−6.5) and quartz sand (3:1). Pots contained 100 g of soil. There was one plant per pot. Every treatment was in three replicates. The maize seeds (Silvestre cv., Producer: KWS Saat, Einbeck, Germany) were seed dressed.

The extracts of plant seeds were obtained by the supercritical $CO_2$ extraction (Łukasiewicz Research Network—New Chemical Syntheses Institute, Puławy, Poland), and the fermentation broths were produced by *Enterobacter* and *Paenibacillus* bacteria in a bioreactor (1L) in the laboratory scale (Bioprocess Laboratory, Grupa Azoty Zakłady Azotowe "Puławy" S.A., Puławy, Poland). In pot experiments with maize, there was an applied liquid nitrogen fertilizer called urea-ammonium nitrate solution with sulphur (UAN+S 28N+5S) (Producer: Grupa Azoty Zakłady Azotowe "Puławy" S.A., Puławy, Poland).

Liquid fertilizer and natural origin products (Table 1) were applied on soil surface in pots after sowing the maize seeds. The biocomponents were used without and with liquid nitrogen fertilizer enriched with sulphur in the following treatments (Table 2):

**Table 2.** Descriptions of the fertilizer treatments in pot experiments.

| Treatments | Descriptions of the Variants |
|---|---|
| $T_c$ | a negative control without biocomponent and UAN+S |
| $T_F$ | a positive control with 10 mL of UAN+S at a total dose of 6 mg N/100 g of soil applied as two split doses (50% of the total dose before sowing and 50% after two weeks of experiment) |
| $T_1$ | E3 at the dose of 10 mL 1%/100 g of soil applied at two split doses |
| $T_2$ | E3 at the dose of 10 mL 1%/100 g of soil and UAN+S at the total dose of 6 mg N/100 g of soil applied at two split doses |
| $T_3$ | E9 at the dose of 10 mL 1%/100 g of soil applied at two split doses |
| $T_4$ | E9 at the dose of 10 mL 1%/100 g of soil and UAN+S at the total dose of 6 mg N/100 g of soil applied at two split doses |
| $T_5$ | S1 at the dose of 10 mL 1%/100 g of soil applied at two split doses |
| $T_6$ | S1 at the dose of 10 mL 1%/100 g of soil and UAN+S at the total dose of 6 mg N/100 g of soil applied at two split doses |
| $T_7$ | S2 at the dose of 10 mL 1%/100 g of soil applied at two split doses |
| $T_8$ | S2 at the dose of 10 mL 1%/100 g of soil and UAN+S at the total dose of 6 mg N/100 g of soil applied at two split doses |

At the end of the experimental growing period (21 days after sowing, BBCH 14–15), the leaf chlorophyll content of the maize plants was measured using a portable non-destructive tool (SPAD—502 Plus, Konica Minolta, Osaka, Japan). Specifically, three SPAD values (SPAD—the Soil Plant Analysis Development) were taken from the base to the apex (along the proximal, central, and distal portions) of the youngest fully expanded leaf of each plant, resulting in a total of 6 measurements (2 plants × 3 repeats) per treatment, and were averaged and expressed as a SPAD index.

At the end of each treatment (after 21 d of plant growth), the seedling length (the distance from soil surface to the upper end of the longest leaves) of the maize cultivar was measured (cm/plant). Subsequently, the maize plants were harvested by separating shoots from roots.

Maize roots were washed in water and then scanned on an Epson root scanner at 400 dpi resolution. The root parameters were analyzed with WIN-RHIZO Arabidopsis 2020 software (Regent Instruments Inc., Québec, QC, Canada).

2.1.3. Effects of Natural Origin Products on In Vitro Fungal Growth

The research material consisted of microbial-origin biocomponents and plant extracts, (Table 1) and pathogenic fungi isolated from corn cobs (*F. culmorum*, *F. fujikuori*, *F. graminearum*). Fungal isolates with the highest pathogenicity were selected in greenhouse tests for the study. Natural origin products were added to sterile potato-dectrose-agar (PDA), medium cooled to 45 °C. in 1, 5, and 10 ppm concentrations. The mixtures of medium and substance were poured into Petri dishes. Agar discs with a diameter of 4 mm, overgrown with the mycelium of individual species of fungi, were placed on the solidified medium in their central part. The plates were incubated at 20 °C under controlled Binder conditions. The measurement of the diameter of the cultures in each combination was made after the mycelial surface had been overgrown in the untreated (control) combination. The average growth of mycelium in millimeters was measured, and then the percentage of mycelial growth inhibition was calculated from the formula:

$$Ow = (K - F/K)\ 100$$

where K is the growth of the fungus on the untreated plates, and F is the growth of the fungus on the plate with the addition of the tested products. The experiment was carried out twice, each time in ten repetitions.

*2.2. Statistical Analysis*

The experimental data of pot experiments were subjected to the analysis of variance using ANOVA analysis (STATISTICA PL). In pots experiments, factor A (conditions) has two levels, and factor B (fertilizer treatments) has ten levels.

Effects of Natural Origin Products and Plant Extracts on In Vitro Fungal Growth

The normality of the distributions of the mycelium colony diameter for three pathogens (*Fusarium culmorum*, *Fusarium fujikuroi* and *Fusarium graminearum*) was tested using Shapiro-Wilk's normality test [22]. Two-way analysis of variance (ANOVA) was carried out to determine the effects of preparation and concentration, as well as preparation × concentration interaction on the variability of the mycelium colony diameter for the three pathogens. The mean values and standard deviations of the distributions of the mycelium colony diameter for the three pathogens were calculated. The Fisher's least significant differences (LSDs) were calculated for individual pathogens, and on this basis, homogeneous groups were determined. The relationships of the mycelium colony diameter between the studied pathogens were estimated using Pearson's correlation coefficients. All analyses were conducted using the GenStat v. 22 statistics software.

## 3. Results

*3.1. Vegetation Experiments*

3.1.1. Seed Germination

It is considered that a germination index (GI) of 100% corresponds to control samples, where maize seeds were treated with distilled water. Therefore, only the biocomponent S2 (the supernatant of fermentation broth from the *Paneibacillus* bacteria) at concentrations of 1 and 2% (*v/w*), leading to a GI higher than 100%, was considered to have biostimulant activity. The supernatant S2 at concentrations of 2 and 1% was superior to control (GI = 100%) by 6.8 and 9.7%, respectively (Figure 1). The treatments with biocomponents E3 (extract of plant seeds from the crop of the legume family (*Fabaceae*), E9 (extract of plant seeds from the crop of the smartweed family (*Polygonaceae*), and S1 (the supernatant of fermentation broth from the *Enterobacter* bacteria) at all studied concentrations have lower germination index compared to the control.

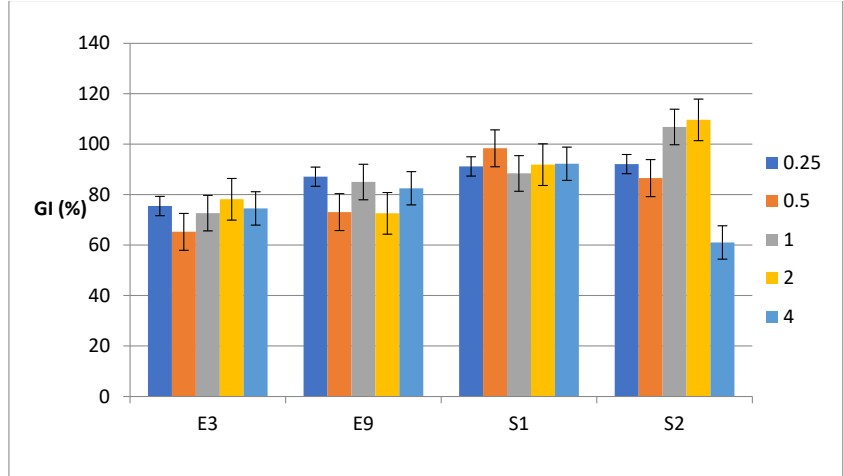

**Figure 1.** Germination index (%), considering the distilled water as the control (100%) for maize seeds grown in Petri dish, treated with four different biocomponents (plant extract E3, plant extract E9, supernatant S1, supernatant S2) in five concentrations (0.25; 0.5; 1; 2; 4%, *w/v*). The bars on the top of the columns represent the SE (*n* = 4).

### 3.1.2. Pot Experiments

In the pot experiments, what was studied were the effects of natural origin products applied alone and with the liquid nitrogen fertilizer, called urea-ammonium nitrate solution with sulphur (UAN+S 28N+5S), on maize plant growth parameters under optimal and drought conditions. Drought affects many aspects of plant growth and development. Our results show that the interaction between two studied factors (condition vs. fertilizer treatment) was not significant in all analyzed parameters of maize seedlings in pot experiments (Table 3). The 7 d exposure to drought stress had a significant influence on maize plant height and root weight in pot experiments. Under drought stress, maize was characterized by the statistically highest plant height compared to optimal conditions in all studied fertilizer treatments. Under optimal conditions, the root weight of maize was statistically highest than in the drought stress conditions in almost all treatments, except treatment T2 (E3, UAN+S), where only the tendency of the higher root weight of maize was observed. The fertilizer treatment (fertilization) had a significant influence on SPAD parameter in maize leaves (Table 3). The statistically highest SPAD in maize leaves was determined when liquid nitrogen fertilizer UAN+S (treatment TF) and plant extract E3 (treatment T1) were applied, compared to supernatant S2 (treatment T7).

**Table 3.** Effects of drought stress and fertilizer treatment on growth parameters of 21-day-old maize seedlings grown in climate chambers.

| Physiological Parameters | Plant Height (cm) | | Stem Weight (g) | | Root Weight (g) | | SPAD [1] | |
|---|---|---|---|---|---|---|---|---|
| Treatment (T) | Optimal | Drought | Optimal | Drought | Optimal | Drought | Optimal | Drought |
| $T_c$ | $37.01 \pm 2.82$ [b] | $47.78 \pm 1.98$ [a] | $1.96 \pm 0.30$ | $2.17 \pm 0.31$ | $3.70 \pm 0.29$ [a] | $2.95 \pm 0.57$ [b] | $33 \pm 1.70$ [A] | $33 \pm 0.87$ [AB] |
| $T_F$ | $42.75 \pm 1.49$ [b] | $47.09 \pm 1.73$ [a] | $2.22 \pm 0.15$ | $1.88 \pm 0.24$ | $4.18 \pm 0.35$ [a] | $2.46 \pm 0.26$ [b] | $35 \pm 1.21$ [A] | $35 \pm 1.33$ [A] |
| $T_1$ | $41.57 \pm 1.93$ [b] | $51.52 \pm 0.25$ [a] | $2.31 \pm 0.01$ | $3.40 \pm 0.07$ | $6.18 \pm 0.24$ [a] | $3.70 \pm 0.63$ [b] | $28 \pm 1.33$ [AB] | $35 \pm 0.92$ [A] |
| $T_2$ | $42.30 \pm 3.97$ [b] | $54.25 \pm 3.75$ [a] | $2.49 \pm 0.39$ | $3.22 \pm 0.66$ | $4.37 \pm 0.08$ | $4.22 \pm 0.39$ | $38 \pm 1.95$ [A] | $34 \pm 1.02$ [AB] |
| $T_3$ | $27.95 \pm 1.95$ [b] | $42.90 \pm 5.10$ [a] | $1.37 \pm 0.05$ | $1.66 \pm 0.30$ | $5.50 \pm 0.42$ [a] | $2.51 \pm 0.74$ [b] | $27 \pm 4.72$ [AB] | $32 \pm 2.87$ [ABC] |
| $T_4$ | $37.00 \pm 1.00$ [b] | $48.90 \pm 0.40$ [a] | $2.24 \pm 0.25$ | $2.22 \pm 0.08$ | $4.60 \pm 0.88$ [a] | $2.44 \pm 0.16$ [b] | $36 \pm 2.13$ [A] | $34 \pm 1.07$ [AB] |
| $T_5$ | $35.50 \pm 1.50$ [b] | $46.45 \pm 2.45$ [a] | $1.54 \pm 0.20$ | $1.75 \pm 0.11$ | $5.17 \pm 0.83$ [a] | $3.46 \pm 0.14$ [b] | $32 \pm 5.43$ [AB] | $31 \pm 0.67$ [ABC] |
| $T_6$ | $41.65 \pm 2.15$ [b] | $47.90 \pm 4.90$ [a] | $2.32 \pm 0.48$ | $2.02 \pm 0.36$ | $5.26 \pm 1.25$ [a] | $2.58 \pm 0.28$ [b] | $37 \pm 0.62$ [A] | $34 \pm 1.12$ [AB] |
| $T_7$ | $39.55 \pm 0.05$ [b] | $45.40 \pm 4.0$ [a] | $2.06 \pm 0.29$ | $1.62 \pm 0.46$ | $5.44 \pm 0.75$ [a] | $2.52 \pm 0.09$ [b] | $21 \pm 2.85$ [B] | $24 \pm 0.37$ [C] |
| $T_8$ | $37.15 \pm 1.95$ [b] | $49.05 \pm 2.60$ [a] | $1.65 \pm 0.22$ | $2.07 \pm 0.10$ | $3.46 \pm 0.98$ [a] | $2.71 \pm 0.20$ [b] | $19 \pm 2.49$ [B] | $26 \pm 5.11$ [BC] |
| LSD ($\alpha = 0.05$): | | | | | | | | |
| Condition (C) | 0.001 | | n.s. | | 0.001 | | n.s. | |
| Treatment (T) | n.s. | | n.s. | | n.s. | | 0.001 | |
| C × T | n.s. | | n.s. | | n.s. | | n.s. | |

[1] SPAD, soil plant analysis development value (chlorophyll content); LSD, least significant difference (statistics); different letters in the same line indicate statistically significant differences, n.s., not significant. Note: different uppercase letters in the same column indicate significant differences among fertilizer treatments ($p < 0.05$); different lowercase letters in the same rows indicate significant differences between optimal and drought conditions ($p < 0.05$).

Moreover, results show that conditions during the vegetation period had a significant influence on root surface area (SA) and root volume (Vol) of maize seedlings (Table 4). The statistically highest SA and Vol parameters were stated under optimal conditions compared to the drought one in almost all studied treatments, except for the treatments of T5 (S1) and T8 (S2, UAN+S), where the SA and Vol parameters were statistically higher in the drought conditions.

**Table 4.** Effects of drought stress and fertilizer treatment on root index for maize.

| Root Index [1] | Len (cm) | | SA (cm²) | | PA (cm²) | | Vol (cm³) | | Avg Diam (mm) | |
|---|---|---|---|---|---|---|---|---|---|---|
| Treatment (T) | Optimal | Drought | Optimal | Drought | Optimal | Drought | Optimal | Drought | Optimal | Drought |
| $T_c$ | 274.93 ± 41.41 | 281.68 ± 26.49 $^{AB}$ | 43.25 ± 2.92 $^{ABCa}$ | 40.53 ± 2.50 $^{ABb}$ | 30.35 ± 0.67 | 26.22 ± 0.80 $^{BC}$ | 0.78 ± 0.06 $^{Ba}$ | 0.69 ± 0.08 $^{BCb}$ | 0.79 ± 0.25 | 0.65 ± 0.03 |
| $T_F$ | 287.24 ± 20.27 | 310.52 ± 32.44 $^{AB}$ | 56.31 ± 5.30 $^{ABCa}$ | 41.96 ± 4.42 $^{ABb}$ | 30.72 ± 1.69 | 29.65 ± 1.44 $^{BC}$ | 1.26 ± 0.24 $^{ABa}$ | 0.66 ± 0.08 $^{BCb}$ | 0.83 ± 0.08 | 0.67 ± 0.07 |
| $T_1$ | 306.10 ± 6.85 | 318.05 ± 17.68 $^{AB}$ | 17.81 ± 0.68 $^{Ca}$ | 16.93 ± 0.71 $^{Bb}$ | 55.94 ± 2.13 | 53.17 ± 2.21 $^{B}$ | 0.59 ± 0.04 $^{Ba}$ | 0.54 ± 0.06 $^{Cb}$ | 0.82 ± 0.08 | 0.72 ± 0.09 |
| $T_2$ | 395.94 ± 92.32 | 450.05 ± 87.37 $^{A}$ | 22.42 ± 5.96 $^{C}$ | 23.12 ± 3.11 $^{AB}$ | 70.43 ± 18.72 | 72.64 ± 9.76 $^{A}$ | 0.57 ± 0.03 $^{Ba}$ | 0.52 ± 0.03 $^{Cb}$ | 1.00 ± 0.30 | 0.94 ± 0.07 |
| $T_3$ | 287.18 ± 26.71 | 234.82 ± 41.22 $^{AB}$ | 75.67 ± 7.10 $^{ABa}$ | 41.26 ± 11.18 $^{ABb}$ | 24.09 ± 2.26 | 13.14 ± 3.79 $^{C}$ | 1.59 ± 0.15 $^{ABa}$ | 0.59 ± 0.23 $^{Cb}$ | 0.84 ± 0.00 | 0.55 ± 0.07 |
| $T_4$ | 253.45 ± 31.07 | 255.95 ± 28.36 $^{AB}$ | 69.11 ± 17.12 $^{ABa}$ | 38.02 ± 2.17 $^{ABb}$ | 22.00 ± 5.45 | 12.10 ± 0.69 $^{C}$ | 1.53 ± 0.56 $^{ABa}$ | 0.45 ± 0.00 $^{Cb}$ | 0.86 ± 0.12 | 0.48 ± 0.03 |
| $T_5$ | 292.05 ± 74.55 | 260.00 ± 74.00 $^{AB}$ | 62.80 ± 13.10 $^{ABb}$ | 67.65 ± 10.25 $^{Aa}$ | 20.00 ± 4.20 | 21.55 ± 3.25 $^{BC}$ | 1.08 ± 0.18 $^{Bb}$ | 1.43 ± 0.02 $^{Aa}$ | 0.70 ± 0.04 | 0.86 ± 0.12 |
| $T_6$ | 257.55 ± 34.85 | 150.00 ± 1.90 $^{B}$ | 71.10 ± 20.10 $^{ABa}$ | 48.10 ± 12.10 $^{ABb}$ | 22.60 ± 6.40 | 15.35 ± 3.85 $^{C}$ | 1.60 ± 0.67 $^{ABa}$ | 1.30 ± 0.60 $^{Ab}$ | 0.86 ± 0.13 | 1.02 ± 0.25 |
| $T_7$ | 268.95 ± 41.94 | 161.70 ± 0.50 $^{B}$ | 80.10 ± 1.95 $^{Aa}$ | 46.25 ± 2.95 $^{ABb}$ | 25.50 ± 0.65 | 14.70 ± 0.95 $^{C}$ | 1.91 ± 0.09 $^{Aa}$ | 1.12 ± 0.11 $^{ABb}$ | 0.94 ± 0.10 | 0.94 ± 0.04 |
| $T_8$ | 232.40 ± 48.45 | 218.35 ± 19.90 $^{AB}$ | 53.20 ± 19.40 $^{ABCb}$ | 61.15 ± 5.95 $^{ABa}$ | 16.95 ± 6.20 | 19.45 ± 1.90 $^{C}$ | 0.98 ± 0.58 $^{Bb}$ | 1.37 ± 0.41 $^{Aa}$ | 0.74 ± 0.06 | 0.89 ± 0.23 |
| LSD ($\alpha$ = 0.05): | | | | | | | | | | |
| Condition (C) | n.s. | | 0.014 | | n.s. | | 0.011 | | n.s. | |
| Treatment (T) | 0.022 | | 0.001 | | 0.004 | | 0.005 | | n.s. | |
| C × T | n.s. | | n.s. | | n.s. | | n.s. | | n.s. | |

[1] Len, root length; SA, root surface area; PA, root projected area; Vol, root volume; AvgDiam, root average diameter; LSD, least significant difference (statistics); different letters in the same line indicate statistically significant differences, n.s., not significant. Note: different uppercase letters in the same column indicate significant differences among fertilizer treatments ($p < 0.05$); different lowercase letters in the same rows indicate significant differences between optimal and drought conditions ($p < 0.05$).

The fertilizer treatments had a significant influence on root length (Len), root projected area (PA), and root volume (Vol) in drought stress condition and on root surface area (SA) in both studied conditions. The statistically highest Len of maize seedlings was stated in treatment T2 (E3, UAN+S), compared to treatments T6 (S1, UAN+S) and T7 (S2) under drought stress. The Len for T2 treatment measured at 450.05 cm, compared to T6 (150.0 cm) and T7 (161.7 cm), and the increase was 200.03 and 178.32%, respectively.

The statistically highest SA of maize seedlings was stated in treatment T7 (S2), compared to T1 (E3) and T2 (E3, UAN+S), under optimal conditions. The SA in T7 was measured 80.10 cm$^2$, compared to T1 (17.81 cm$^2$) and T2 (22.42 cm$^2$), and the increase was 349.24 and 257.27%, respectively. Under drought stress conditions, the statistically highest SA of maize seedlings was stated in treatment T5 (S1), compared to T1 (E3). The SA in T5 was measured at 67.65 cm$^2$, compared to T1 (16.93 cm$^2$), and the increase was 299.59%. The statistically highest PA was stated in treatment T2 (E3, UAN+S), and the lowest in the following treatments: T3 (E9), T4 (E9, UAN+S), T6 (S1, UAN+S), T7(S2), and T8 (S2, UAN+S) under drought stress conditions. The PA in T2 was measured at 72.64 cm$^2$, compared to T3 (17.81 cm$^2$), T4 (12.10 cm$^2$), T6 (15.35 cm$^2$, T7 (14.7 cm$^2$), and T8 (19.45 cm$^2$), and the increase was from 273.47 to 500.33%.

Under optimal conditions, the statistically highest Vol was stated in treatment T7, compared to the following treatments: $T_C$, T1, T2, and T5. The Vol in T7 (1.91 cm$^3$) was measured and compared to the lowest Vol values obtained in the above listed treatments, and the increase was from 76.85 to 235.09%.

The statistically highest Vol was stated in treatments T5 (S1), T6 (S1, UAN+S), and T8 (S2, UAN+S), and the lowest in the following treatments: T1 (E3), T2 (E3, UAN+S), T3 (E9), and T4 (E9, UAN+S), under drought stress conditions. The Vol values for T5, T8, and T6 measured 1.43, 1.37, and 1.30 cm$^3$, respectively. Comparing the results of measured Vol to the lowest Vol value in, e.g., T4 (0.45 cm$^3$), the increase was in the range of 188.88–222.22% (Table 4).

### 3.1.3. Effects of Natural Origin Products on In Vitro Fungal Growth

Differences in mycelial growth were found after using plant extracts and microbial origin biocomponents products.

Analysis of variance indicated that the main effect of tested biostimulants was significant for the mycelium colony diameter for all three pathogens (Table 5). The main effect of concentration was significant only for the mycelium colony diameter of *F. culmorum*. Preparation × concentration interaction was significant for *F. fujikuroi* and *F. graminearum*.

**Table 5.** Mean squares from two-way analysis of variance for the mycelium colony diameter for the three pathogens.

| Source of Variation | d.f. | *F. culmorum* | *F. fujikuroi* | *F. graminearum* |
|---|---|---|---|---|
| Preparation | 4 | 8650.93 *** | 9463.54 *** | 13,241.68 *** |
| Concentration | 2 | 223.88 * | 172.31 | 77.7 |
| Preparation × Concentration | 8 | 104.5 | 270.42 ** | 112.87 ** |
| Residual | 75 | 52.54 | 78.71 | 33.22 |

* $p < 0.05$; ** $p < 0.01$; *** $p < 0.001$; d.f.—the number of degrees of freedom.

The use of plant extracts and microbial origin biocomponent S1 in all tested concentrations (except extract of a plant from the leguminous family at a concentration of 10 ppm—E3) did not statistically significantly reduce the diameter of growth of *F. culmorum* colonies (Table 6). Only the S2 microbial origin biocomponent applied in all tested concentrations significantly limited the growth of fungal colonies. Mycelial growth of *F. fujikuori* was statistically significantly reduced after the application of both microbial origin biocomponents (S1 and S2) at concentrations of 1, 5 and 10 ppm. Application of a lower concentration of the S2 biocomponent resulted in a higher reduction of the growth of mycelium. Application of the S2 microbial origin biocomponent at all tested concentrations,

and application of the S1 biocomponent at a concentration of 1 ppm, had a limiting effect on the growth of *F. graminearum.* Additionally, the use of the plant origin extract E9 at a concentration of 5 ppm significantly reduced the growth of the mycelium of mentioned-above fungi. Similar results as for *F. fujikuori* were obtained after the application of the S2 microbial origin biocomponent. An inverse relationship was noted in limiting the growth of mycelium—an increase in the concentration of the biocomponents resulted in a higher growth of mycelium. A similar trend was noted for the S1 biocomponent. Application of the S1 microbial origin biocomponent resulted in obtaining the smallest diameter of the mycallium, *F. fujikuori* and *F. graminearum*, at the lowest used concentration—1 ppm, which was 32.2 mm and 19.8 mm, respectively (Table 6).

**Table 6.** Mean values of the mycelium colony diameter for all three pathogens.

| Patogen | *Fusarium culmorum* | | | | *Fusarium fujikuroi* | | | | *Fusarium graminearum* | | | |
|---|---|---|---|---|---|---|---|---|---|---|---|---|
| Concentration (C) [%] | 1 | 5 | 10 | Mean | 1 | 5 | 10 | Mean | 1 | 5 | 10 | Mean |
| Preparation (P) | Diameter of Mycellium [mm] | | | | Diameter of Mycellium [mm] | | | | Diameter of Mycellium [mm] | | | |
| Control | 90.0 | 90.0 | 90.0 | 90.0 a | 90.0 | 90.0 | 90.0 | 90.0 a | 90.0 | 90.0 | 90.0 | 90.0 a |
| E3 | 88.8 | 86.3 | 79.0 | 84.7 b | 84.3 | 85.8 | 84.5 | 84.9 a | 90.0 | 90.0 | 90.0 | 90.0 a |
| E9 | 90.0 | 90.0 | 90.0 | 90.0 a | 90.0 | 90.0 | 90.0 | 90.0 a | 88.7 | 81.2 | 89.0 | 86.3 b |
| S1 | 90.0 | 89.0 | 90.0 | 89.7 a | 60.5 | 41.8 | 66.5 | 52.3 b | 83.5 | 87.3 | 90.0 | 86.9 ab |
| S2 | 44.3 | 45.7 | 29.5 | 39.8 c | 32.2 | 43.7 | 43.3 | 39.7 c | 19.8 | 35.0 | 28.5 | 27. 8c |
| Mean | 80.6 A | 80.2 A | 75.7 B | | 71.4 AB | 70.3 B | 74.9 A | | 74.4 B | 76.7 AB | 77.5 A | |
| LSD$_{0\cdot05}$ | Preparation: 4.813; Concentration: 3.728; P × C: 8.337 | | | | Preparation: 5.891; Concentration: 4.563; P × C: 10.204 | | | | Preparation: 3.827; Concentration: 2.964; P × C: 6.629 | | | |

a, b, c—In columns, means followed by the same letters are not significantly different. A, B—In rows, means followed by the same letters are not significantly different.

## 4. Discussion

At low concentrations, biostimulants improve germination, whereas in high concentrations, an inhibitory effect is often observed on seed germination. These effects are also plant species–dependent. In our research, only the microbial origin biocomponent S2 (the supernatant of fermentation broth from *Paenibacillus* bacteria), at 1 and 2% concentrations, had the positive influence on the germination index of maize seeds compared to the control (distilled water), and therefore is considered to have biostimulant activity. In the case of the plant seed extracts from the crop of the legume family (*Fabaceae*) and from the crop of the smartweed family (*Polygonaceae*), at studied concentrations, they did not have positive effects on GI. According to the other research [23], the use of increasing doses of the biostimulant provides increases in seedling vigor, up to the dose of 0.50 L per 100 kg of seeds. Thus, doses of up to 0.50 L per 100 kg of seeds provide a rapid and uniform emergence, allowing greater development of the adventitious roots and favoring the absorption of nutrients in the period between emergence and evaluation. Moreover, the excess of nutrients and phytohormones can cause toxicity to the plant, affecting the cellular metabolism and the initial development of seedlings, besides reducing their vigor [23].

Recent studies have projected that by 2050, the global average temperature will rise and probably exceed by 2 °C under the current high emission scenario. It will cause an additional maize yield loss of 10 million tons per year, with increasing temperature and changing rainfall patterns [24]. Hence, the novel biostimulants can play a significant role in the manipulation of maize growth traits under drought stress. Drought stress affects growth rates during the vegetative stage of maize by lowering the active photosynthetic leaf area of the crop canopy. The first response to stress is turgor loss that decreases the growth rate, stem elongation, foliar expansion, and stomatal opening [25]. The fastest response to water deficit is stomatal closure to protect the plant from water loss. Water

deficit produces abscisic acid (ABA) biosynthesis, which triggers stomatal closure and causes a decrease in intracellular $CO_2$ concentration and photosynthesis inhibition. Polyols, such as mannitol, quaternary ammonium salts, such as glycine betaine, amino acids, such as proline, and sugars, such as trehalose, are solutes that are compatible with a metabolism that can accumulate and play an important role in maintaining cellular turgor, and protect membranes and proteins from irreversible damage caused by water loss [26].

The effects of promoting crop growth and reducing stress symptoms (e.g., water stress, unfavorable temperatures, nutrient deficit, etc.) depend on several factors, including the timing and type of biostimulant and/or fertilizer application, and stress intensity. Some research showed that the studied biostimulant (e.g., ComCat®), applied with and after crops had been exposed to stress, did not promote maize biomass production under favorable growing conditions, regardless if it was applied at low or high rates [27]. In the review [28], it is pointed out that biostimulants might have to be applied before the stress occurs. If biostimulants could be provided with a detailed label describing the proper timing and rate of application and the mode of action in different crops, their practical use could be improved, and the waste of product could be avoided [28].

In our study, we studied the effects of natural origin products (plant and microbial origin biocomponents) applied alone and with the liquid nitrogen fertilizer, called urea-ammonium nitrate solution with sulphur (UAN+S 28N+5S), on maize plant growth parameters under optimal and drought conditions in pot experiments. Nitrogen (N) fertilization is one of the most important agrotechnical treatments that enables the farmer to obtain desired crop yields. Treatments aiming at increasing yields of cereals, including maize, must focus on a more efficient use of nitrogen contained in mineral fertilizers. Sulphur (S) plays an important role in the formation of chlorophyll and biosynthesis of proteins and lipids in plants. Furthermore, good S supply has a positive influence on the uptake of other nutrients and efficiency of fertilization. Sulphur is also beneficial for the plant growth parameters, yield structure elements, and consequently, for the yields of maize [29,30]. Hence, this study was undertaken to test the effects of fertilization of maize with a solution of urea and ammonium nitrate (UAN) enriched with sulfur, and with different natural origin biocomponents applied alone or in combinations with the studied liquid fertilizer. In the experiment, the first doses of biocomponents and liquid nitrogen fertilizer were applied before the occurrence of drought stress. The natural origin products showed a positive response for root weight in optimal conditions. The negative control, without any biocomponent and fertilizer, showed better performance in drought conditions than optimal conditions in the case of plant height of maize. The optimal temperature for maize growth is 21–27 °C. It is the plant of type C4. Above 32 °C, the plant grows more slowly, and the yield is reduced. Under drought stress, the temperature was 30/21 °C (photoperiod 16/8 h), and in optimal conditions, the temperature was 21/16 °C (photoperiod 16/8 h). Moreover, in drought conditions, the root index for maize (SA and Vol) was the statistically highest in treatments T5 (S1) and T8 (S2, UAN+S). In optimal conditions, the root index for maize (SA and Vol) was statistically highest in treatment T7 (S2). Therefore, it can be concluded that microbial origin biocomponents have positive effects on the root parameters, and reduce stress symptoms under water deficit.

Among the tested products, the microbial origin extracts S2, based on fermentation broth (supernatant 2) obtained with the use of bacteria from the genus *Paenibacillus*, was the most effective in limiting the growth of mycelium of fungi. The applied products of plant origin—E3 and E9—inhibited the development of mycelium of fungi of the genus *Fusarium* in a small percentage. In the in vitro studies by Salhi et al. [31], the authors examined the influence of the plants *Artemisia herba alba*, *Cotula cinerea*, *Asphdelus tenuifolius*, and *Euphorbia guyoniana* (growing in the natural environment) extracts on the *F. graminearum* and *F. sprotorichioides* mycelium growth. After using all of the tested plant extracts applied in concentrations of 10 and 20%, researchers noted a higher inhibition of mycelial growth compared to the plant extracts tested in the authors' experiments. Keriene et al. [32] studied the ability of buckwheat hull extract in inhibiting mycelial growth of *Fusarium culmorum*

and *F. graminearum*. Buckwheat, like the E9 plant extract tested in the authors' experiments, is a plant from the *Polygonacea* family. The highest antifungal properties of the extracts were observed when grains had been exposed to them for the longest time; *Fusarium* spp. growth on buckwheat grain was highly inhibited when the exposure time was 90 min at a 25 °C temperature. In the authors' research, an extract from a plant from the *Polygonacea* family reduced the mycelium growth of the fungus *Fusarium graminearum* only in one of the tested concentrations—5 ppm.

Satish et al. [33] tested 46 plant extracts belonging to 32 families, including two extracts belonging to the *Fabacae* family. In the research mentioned above, extracts were tested in reducing mycelium growth of eight species of fungi of the genus *Fusarium* found on maize, paddy and sorghum seeds. The authors did not find any antifungal activity of extracts from the *Fabacae* family, and obtained similar effects to those observed in the Satish et al. research. Fungal activity of plant extracts of *Ammi visnaga*, *Eucalyptus globulus*, *Artemisia judaica*, and *Coriandrum sativum* on *F. fujikuroi* was tested in vitro (in higher concentrations from 250 to 1250 ppm) on linear growth of *F. fujikuroi* on rice by Kalboush and Hassan [34]. The best effect was obtained by increasing the concentrations of plant extracts. In our own research, in the case of the microbial origin biocomponent (S2), an increase in inhibition of *F. fujikuori* mycelium with an increase of concentration was not observed. Research on antifungal activity against *Fusarium* pathogen's occurrence on maize was also conducted by Seepe et al. [35]. In studies with extracts of eight medicinal plants, 97% inhibition of *F. proliferatum* mycelium growth was found after using *Melia azederach* extract. In our own experiments, such a significant inhibition of mycelial growth was not observed; however, the results obtained after adding a microbial origin biocomponent (S2), based on fermentation broth (supernatant 2), were satisfactory and will be continued in greenhouse conditions and field.

## 5. Conclusions

The biocomponent S2 of microbial origin (supernatant 2) obtained with the use of bacteria of the genus *Paenibacillus* has potential for use in seed treatment as it provides better initial development of maize seedlings (germination index) compared to other studied biocomponents (E3, E9, and S1). The different conditions (optimal or drought) in the pot experiments had a significant influence on some physiological (e.g., plant height and root weight) and root parameters (e.g., SA and Vol) of maize seedlings. The type of fertilizer treatment had a significant influence on the SPAD parameter in maize leaves and on almost all studied root parameters of maize seedlings (e.g., Len, SA, PA, Vol).

Among the tested plant extracts and microbial products used to limit mycelial growth, only the biocomponent S2 of microbial origin showed a mycelial growth limiting effect on all tested *Fusarium* fungi isolated from corn cobs. The microbial origin biocomponents S2 inhibited the growth of *F. fujikuori* and *F. graminearum* better at a lower concentration, while at a higher concentration it limited the growth of *F. culmorum* mycelium growth. The second tested microbial origin biocomponent, S1, inhibited the growth of *F. fujikuori* mycelium at all tested concentrations and *F. graminearum* at the lowest tested concentration. Among the tested plant extracts, only a reduction in the growth of *F. graminearum* mycelium was observed after adding the E9 extract to the medium at a concentration of 5 ppm.

Nowadays, modern agriculture needs to review and broaden its practices and business models by integrating opportunities coming from different adjacent sectors and value chains, including the bio-based industry, in a fully circular economic strategy. Searching for new tools and technologies to increase crop productivity under optimal and sub-optimal conditions, and to improve resource use efficiency, is crucial to ensure food security while preserving soil quality, microbial biodiversity, and providing business opportunities for farmers. This study demonstrated that the use of natural origin biocomponents, especially microbial origin products, help to alleviate drought stress, and can reduce some diseases in maize plants. There is a need for more research to provide guidance to farmers on which biostimulants proves more beneficial to specific crops, and to understand the influence of natural origin biocomponents on plant physiology.

**Author Contributions:** Conceptualization, M.M.-S. and J.H.; methodology, M.M.-S., J.D., E.J. and M.K. (Marta Klimczyk); software, J.B.; validation, M.M.-S., J.D., E.J., M.K. (Marek Korbasand) and L.S.; formal analysis, M.M.-S., J.D., E.J. and M.K. (Marta Klimczyk); investigation, M.M.-S.; resources, M.M.-S.; writing—original draft preparation, M.M.-S., J.D., E.J., M.K. (Marek Korbasand), L.S., M.K. (Marta Klimczyk) and J.B.; writing—review and editing, M.M.-S., J.D., E.J., M.K. (Marek Korbasand), L.S., M.K. (Marta Klimczyk) and J.B.; visualization, J.H., J.D., E.J. and M.K. (Marek Korbasand); supervision, M.M.-S., J.H. and M.K. (Marek Korbasand); project administration, M.M.-S. and M.K. (Marek Korbasand); funding acquisition, M.M.-S. and M.K. (Marek Korbasand), J.D. and J.H. All authors have read and agreed to the published version of the manuscript.

**Funding:** This research was funded by The National Centre for Research and Development (NCBR), grant number POIR.01.01.01-00-1265/20.

**Institutional Review Board Statement:** Not applicable.

**Data Availability Statement:** Not applicable.

**Acknowledgments:** We thank Marcin Podleśny, Monika Szymajda, Jagoda Kucharska, and Tomasz Szymczak (Grupa Azoty Zakłady Azotowe "Puławy" S.A., The Bioprocess Laboratory) for the selection of microorganisms, technical support, and the production of the microbial origin products. We thank Marta Wyzińska (The Institute of Soil Science and Plant Cultivation—State Research Institute, Puławy, Poland) for sharing the maize seeds used in the experiments.

**Conflicts of Interest:** The authors declare no conflict of interest.

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
