# Peer review of "The Assessment of an Effect of Natural Origin Products on the Initial Growth and Development of Maize under Drought Stress and the Occurrence of Selected Pathogens"

_agriculture, doi:10.3390/agriculture13040815_

Round 1

Reviewer 1 Report

1.      The novelty of this work is not clear comparing with existing works discussed in Section 1.

2.      The main contributions of this manuscript are not summarized clearly in the manuscript. The authors should also add discussion about differences between the proposed work and related works.

3.      The contents in this manuscript are organized poorly. There are lots of redundant irrelevant details in the manuscript.

4.      There are a some of grammar mistakes, typing errors in the manuscript.

Author Response

Response to Reviewer 1 Comments

  1.     The novelty of this work is not clear comparing with existing works discussed in Section 1.
  2.     The main contributions of this manuscript are not summarized clearly in the manuscript. The authors should also add discussion about differences between the proposed work and related works.
  3.     The contents in this manuscript are organized poorly. There are lots of redundant irrelevant details in the manuscript.
  4.     There are a some of grammar mistakes, typing errors in the manuscript.

Answer: Dear Reviewer! We kindly agree with the mentioned missings and comments. Paper is now changed and the sections of the article are now improved. Missing data is clearly organized in then tables, grammar check was managed. Introduction, methodology and discussion sections are now upgraded with the necessary information. We hope now it meets Your requirements. Authors.

Reviewer 2 Report

The assessment of an effect of natural origin products on the initial growth and development of maize under drought stress and the occurrence of selected pests

Manuscript Number: Agriculture-2273910

In my opinion, the subject matter dealt with by the authors is very interesting, because of boosting up plant growth using natural products under drought stress is great idea. However, having thoroughly reviewed the manuscript presented to me, I have some major comments and suggestions, which I present below:

1.      The manuscript suffers from lots of grammatical and spelling mistakes.

2.      The properties of soil used in this study is missing.

3.      The extract preparation part is missing which part was extracted also missing.

4.      Whether the extract was prepared from the weed or crop of legume family didn't mention here. Because weed extract has more negative allelopathic effect on seed germination and plant growth that is mentioned result section.

5.      Weed extract at very high concentration show a substantial amount negative effect on germination. Here, 4% concentration is high enough to inhibit germination. But it is not showing in the result.

6.      Enterobacter also produce IAA, GA3, ZA, iron carrier and ACC deaminase that have the capability to influence germination and growth of plant as Paenibacillus. But there was no positive or negative effect was found for fermentation broth.

7.      Why physiological parameters values are enough high in drought stress than in optimal condition even all the treatments were applied in optimal condition. What does that mean? Why does not the natural origin product show any positive response in optimal condition?

8.      Why a negative control without any biocomponent and fertilizer showed better performance in drought condition than optimal condition?

Author Response

Response to Reviewer 2 Comments

In my opinion, the subject matter dealt with by the authors is very interesting, because of boosting up plant growth using natural products under drought stress is great idea. However, having thoroughly reviewed the manuscript presented to me, I have some major comments and suggestions, which I present below:

  1. The manuscript suffers from lots of grammatical and spelling mistakes.

Answer: We kindly agree with the Reviewer’s comment. Grammar and spelling mistakes were corrected and the paper was rechecked once again.

  1. The properties of soil used in this study is missing.

Answer: Properties of soil are added in material and methods section. Now it should fullfill the Reviewer’s request.

  1. The extract preparation part is missing which part was extracted also missing.

Answer: Method of extract production was added to the text (supercritical CO2 extraction of plant seeds).

  1. Whether the extract was prepared from the weed or crop of legume family didn't mention here. Because weed extract has more negative allelopathic effect on seed germination and plant growth that is mentioned result section.

Answer: The extract was made from crop of legume family not weed. Information is now added in the text.

  1. Weed extract at very high concentration show a substantial amount negative effect on germination. Here, 4% concentration is high enough to inhibit germination. But it is not showing in the result.

Answer: Authors study results didn’t show high germination inhibition at 4% in the case of studied extracts.

  1. Enterobacter also produce IAA, GA3, ZA, iron carrier and ACC deaminase that have the capability to influence germination and growth of plant as Paenibacillus. But there was no positive or negative effect was found for fermentation broth.

Answer: Its true, in our results of the experiment with maize Authors haven’t observed the similar effect in case of this two types of fermenation broths but in experiments with wheat the positive effects was observed for both Enterobacter and Paenibacillus on wheat growth (data not shown in this article). Moreover, the type of metabolites which are produced are dependent on fermentation parameters.

  1. Why physiological parameters values are enough high in drought stress than in optimal condition even all the treatments were applied in optimal condition. What does that mean? Why does not the natural origin product show any positive response in optimal condition?

Answer: The observed dependency could be connected with the short duration of the experiment. It was not possible to extend the time of experiments from technical point of view because of the height of maize plants in climatic chamber and their whithering.

  1. Why a negative control without any biocomponent and fertilizer showed better performance in drought condition than optimal condition?

Answer: The negative control without any biocomponent and fertilizer showed better performance in drought conditions than optimal conditions only in case of plant height. The optimal temperature for maize growth is 21-27oC. It is a plant of type C4. Above 32oC, the plant grows more slowly and the yield is reduced. In drought stress the conditions in climate chambers were: photoperiod 16/8h, temperature 30/21°C and in optimal: photoperiod 16/8h, temperature 21/16°C. The conditions were settled universal for three plants that were in project to be able compare the results.

Reviewer 3 Report

Here are my comments

Abstract
Seventy five percent part of the abstract is based on introductory sentences. Methodology is completely missing and results in abstract are not showing any numerical value.

Introduction
At line 69 directly discussing the bio-stimulant. Need few lines about the nature of bio-stimulant is required. aAs bio-stimulant is highly generalized term.

Methodology. Its quite confusing for readers. 
In first experiemnt how did you get the extract of  both plants. The methodology is missing. How it was applied.? 

urea-ammonium nitrate solution with sulphur was also used but is it part of bio stimulant? When was water stress  applied? How the effect of water stress was measured? At what stage the samples were inoculated with fungus? Nothing is clear here in methodology. 
Detailed chemcial analysis of your palnt extract is also required. 
After adding all this paper can be reviewed again.

Author Response

Response to Reviewer 3 Comments

Here are my comments

Abstract
Seventy five percent part of the abstract is based on introductory sentences. Methodology is completely missing and results in abstract are not showing any numerical value.

Answer: We kindly agree with the Reviewer’s comment. The Abstract is corrected and changes were made to improve this section of the paper.

Introduction
At line 69 directly discussing the bio-stimulant. Need few lines about the nature of bio-stimulant is required. aAs bio-stimulant is highly generalized term.

Answer: We kindly agree with the Reviever’s comments introduction section is now improved. Biostimulant nature is now extended in the text.

Methodology

Its quite confusing for readers. 
In first experiemnt how did you get the extract of  both plants. The methodology is missing. How it was applied.? 

urea-ammonium nitrate solution with sulphur was also used but is it part of bio stimulant? When was water stress  applied? How the effect of water stress was measured? At what stage the samples were inoculated with fungus? Nothing is clear here in methodology. 

Answer: UAN+S was first applied using pippete and then biostimulants, both on the soil surface of pots after sowing the seeds. Stress was applied after two weeks and lasted 7 days. The plants were not watered additionally to the initial weight of pots. Experiments with fungus were done by other institution (IOR) and this is not connected with pot experiments.

Detailed chemcial analysis of your palnt extract is also required. 
After adding all this paper can be reviewed again.

Answer: Detailed chemical analysis of plant extracts isn’t done yet and from the point of view of intellectual property protection, the entrepreneur does not want to disclose it.

Reviewer 4 Report

General recommendations and questions

Title

None of the sections of the article were devoted to selected pests. The title should be corrected.

Abstract

The introductory part of the Abstract is too long, it is recommended to shorten it.

“Results showed that microbiological extracts: S1 and S2 had a positive effect on the germination index (GI> 100%) of maize seeds compared to plant extracts (E3, E9) (GI <100%). The extracts E3, S1 and S2 used as a single product and in combination with UAN+S, under optimal conditions…..”For the reader, such abbreviated designations do not mean anything, please give a brief description of the studied preparations.

Introduction

Line 52. “per land unit area” – do You mean unit of land area?

Line 57. Water instead of weater.

Line 86-87. “The following are used for this purpose: extracts, plant origin extracts, vegetable oils [20]” What kind of extracts are we talking about here?

It would be advisable to mention the tasks to achieve the goal, in which, at least in general terms, the products to be tested are described.

Materials and Methods

2.1.1. Seed germination

Were the same products from Table 1 used for seed germination studies? If so, this should be indicated and the products should be described.

2.1.2. Pot experiments

When you first mention, give full names: what is RH, C.

I would recommend giving the descriptions of the variants in the form of a Table, otherwise it is very difficult to follow the idea and the results would be easier to understand later.

Line 130. What is “Soil Plant Analysis Development” – is that the name of the devise?

Nothing is described about the pot experiment - what kind of substrate, how big pots, how many plants in a pot, how many pot per variants, etc.

Where did you take the preparations under study - are they bought, prepared yourself, generally available? What extracts - water, ethanol...?

Why did you choose this fertilizer? Justify a little.

  2.1. 3. Did you prepare fungal isolates yourself? How did you choose the most pathogenic ones? Are they commercially available, experimentally obtained, described somewhere before? If there are no references or descriptions, I have to take your word for it, but it is not scientifically correct.

2.2.2. If the full Latin name for the species has already been given once, the abbreviated form can be used further in the text. Why were other pathogens used for the statistical analysis than those described in methods section 2.1.3.

Results

Seed germination.

Why is there no statistical analysis of data for the germination experiment? Why is the control not reflected in Fig.1? If the standard errors have already been determined, why has it not been evaluated whether there are significant differences between the variants. How can you tell if something is better than a control if the control is not specified?

Pot experiments

You are speaking about two factors – moisture conditions and fertilization treatments. Why has the effect of the preparations not been analysed? Why only fertilization? Or is it an unfortunate form of expression?

Table 2. What do you mean with - different letters in the same line indicate statistically significant differences? If you compare results in the line - it refers to the conditions, if in the column - to the studied treatments. For correct data analysis, I recommend using both upper and lower case letters - one for comparing data in columns, the other for rows.

I would also recommend including a brief description of the treatments in the tables. It's inconvenient to go back to the Methods section and it's hard to understand there too.

No reference in text to Table 3. Comments for Table 3 - similar to Table 2.

Line 204-205. The sentence is not understandable.

In general, this section is too long, there is no need to retell the data shown in the tables. Recommendation to pay attention to the main things. You should not jump from optimal to drought conditions and back. Create a logical system for presenting key results.

3.1.3. Effects of natural origin products on in vitro fungal growth

Table 4. Unclear name of the table.

Line 241-242. What does it mean – “the main effect of preparation was significant”?

Table 5 is not mentioned in the text. Why is it not specified, which is significantly different in terms of concentrations and preparations? I recommend using both upper and lower case letters.

“Application of S1 microbial origin extract resulted in obtaining the smallest diameter of the fungi F. fujikuori and F. graminearum” – diameter of fungi or mycelium?

 Discussion

The main disadvantage of Discussion - if you pay too much attention to other studies, you lose focus on your results. So, for example, there are only 2 sentences dedicated to the authors' research in an entire Discussion page. In addition, in the second paragraph, which describes the researches of other authors, the reference [26] are not in the literature list at all.

The results of the pot experiment, to which the largest and most difficult to understand Results section is devoted, have not been discussed at all. Such a single, meaningless sentence cannot be taken seriously: “On some of the studied physiological and root parameters of 21 d maize seedlings in controlled conditions had the influence both type of condition and type of fertilizer treatment.”

Line 285. “According to other research, the use of increasing doses of the biostimulant provide …..” there are no references for other research.

Line 312-314. The sentence is difficult to understand, mainly because of the strange expression.

I would suggest avoiding saying “The Authors research”. Instead - our research, the research results, our experimental results etc.

In general, I would recommend making the Discussion part more focused. More purposefulness in data analysis is needed, linking them with the results of other authors' research. It should be clearly indicated what, according to the authors, would be ineffective, what would be promising, what could be used in practice.

The article is devoted to such a topical topic as the use of plant biostimulants of natural origin to promote plant growth and improve tolerance to a wide range of biotic and abiotic stress factors and is generally interesting. However, there are many inaccuracies in the article. The description of the methodology for the experimental variants and products used can definitely be improved. The Results section is too long and more attention should be paid to the correct statistical analysis of the results. The Discussion part should be more focused.  More purposefulness in data analysis is needed, linking them with the results of other authors' research.  I recommend accepting this article in Plants after major revision. I recommend that efforts be made to improve the scientific quality of this work, as the data material obtained in this study could be of interest to both plant physiology and agricultural scientists.

Author Response

Response to Reviewer 4 Comments

Title

None of the sections of the article were devoted to selected pests. The title should be corrected.

Answer: We kindly agree with the Reviever’s comment. The title is now corrected.

Abstract

The introductory part of the Abstract is too long, it is recommended to shorten it.

“Results showed that microbiological extracts: S1 and S2 had a positive effect on the germination index (GI> 100%) of maize seeds compared to plant extracts (E3, E9) (GI <100%). The extracts E3, S1 and S2 used as a single product and in combination with UAN+S, under optimal conditions…..”For the reader, such abbreviated designations do not mean anything, please give a brief description of the studied preparations.

Answer: We kindly agree with the Reviewer’s comment. The description is now given.

Introduction

Line 52. “per land unit area” – do You mean unit of land area?

Answer: The sentence is now corrected to „unit of land area”.

Line 57. Water instead of weater.

Answer: Sentence is now corrected to „water”

Line 86-87. “The following are used for this purpose: extracts, plant origin extracts, vegetable oils [20]” What kind of extracts are we talking about here?

It would be advisable to mention the tasks to achieve the goal, in which, at least in general terms, the products to be tested are described.

Answer: We kindly agree with the Reviewer’s comment. Extracts from plant seeds and microorganisms metabolites (supernatans). The description is now given in the text.

Materials and Methods

2.1.1. Seed germination

Were the same products from Table 1 used for seed germination studies? If so, this should be indicated and the products should be described.

Answer: The same products in germination and pot tests were tested.

When you first mention, give full names: what is RH, C.

Answer: We kindly agree with the Rewiever’s comment. The abbreviation C nad RH is now corrected in the text.

2.1.2. Pot experiments

I would recommend giving the descriptions of the variants in the form of a Table, otherwise it is very difficult to follow the idea and the results would be easier to understand later.

Answer: We kindly agree with the Rewiever’s comment. The description of variants is now in the table 2.

Line 130. What is “Soil Plant Analysis Development” – is that the name of the devise?

Answer: Line 130 is only explanation of abbrev SPAD (value), but is also in the name of the device. The text is now extended.

Nothing is described about the pot experiment - what kind of substrate, how big pots, how many plants in a pot, how many pot per variants, etc.

Answer: We kindly agree with the Rewiever’s comment. Pot experiments methodology is now corrected: soil, plant numer per pot, replications, etc.

Where did you take the preparations under study - are they bought, prepared yourself, generally available? What extracts - water, ethanol...?

Answer: Extracts were produced by Subcontractor by the supercritical CO2 extraction

Why did you choose this fertilizer? Justify a little.

Answer: In discussion part of the paper we added justification of the fertilizer that we use for experiments.

  2.1. 3. Did you prepare fungal isolates yourself? How did you choose the most pathogenic ones? Are they commercially available, experimentally obtained, described somewhere before? If there are no references or descriptions, I have to take your word for it, but it is not scientifically correct.

Answer: For the experiment Fusarium culmorum, Fusarium fujikuroi and Fusarium graminearum cultures isolated from corn cobs of the highest pathogenicity, selected in greenhouse tests, were used.

2.2.2. If the full Latin name for the species has already been given once, the abbreviated form can be used further in the text. Why were other pathogens used for the statistical analysis than those described in methods section 2.1.3.

Answer: We kindly agree with the Rewiever’s comment. Changes are now applied in the text.

Results

Seed germination.

Why is there no statistical analysis of data for the germination experiment? Why is the control not reflected in Fig.1? If the standard errors have already been determined, why has it not been evaluated whether there are significant differences between the variants. How can you tell if something is better than a control if the control is not specified?

Answer: It is now mentioned in the description of the table. Control for GI=100%

Pot experiments

You are speaking about two factors – moisture conditions and fertilization treatments. Why has the effect of the preparations not been analysed? Why only fertilization? Or is it an unfortunate form of expression?

Answer: It is just an unfortunate form of expressions, the title of table is now corrected

Table 2. What do you mean with - different letters in the same line indicate statistically significant differences? If you compare results in the line - it refers to the conditions, if in the column - to the studied treatments. For correct data analysis, I recommend using both upper and lower case letters - one for comparing data in columns, the other for rows.

Answer: We kindly agree with the Rewiever. We change the letters in tables as you suggested: upper case letters – for columns and lower case letters for rows

I would also recommend including a brief description of the treatments in the tables. It's inconvenient to go back to the Methods section and it's hard to understand there too.

Answer: We change the description of the treatments to the table to make it more transparent for the readers.

No reference in text to Table 3. Comments for Table 3 - similar to Table 2.

Answer: Table 3 now is 4 and is cited in the text.

Line 204-205. The sentence is not understandable.

Answer: Mentioned sentence is now corrected.

In general, this section is too long, there is no need to retell the data shown in the tables. Recommendation to pay attention to the main things. You should not jump from optimal to drought conditions and back. Create a logical system for presenting key results.

Answer: It was corrected to more logical system for presenting results.

3.1.3. Effects of natural origin products on in vitro fungal growth

Table 4. Unclear name of the table.

Line 241-242. What does it mean – “the main effect of preparation was significant”?

Answer:  It is now changed for the “main effect of biostimulants was significant”. We mean the influence of mentioned biostiumulant was significant on the mycelium growth”

Table 5 is not mentioned in the text. Why is it not specified, which is significantly different in terms of concentrations and preparations? I recommend using both upper and lower case letters.

Answer:  It is now corrected. We added "a, b, c - In columns, means followed by the same letters are not significantly different.
A, B - In rows, means followed by the same letters are not significantly different."

“Application of S1 microbial origin extract resulted in obtaining the smallest diameter of the fungi F. fujikuori and F. graminearum” – diameter of fungi or mycelium?

Answer: Mycelium - corrected.

 Discussion

The main disadvantage of Discussion - if you pay too much attention to other studies, you lose focus on your results. So, for example, there are only 2 sentences dedicated to the authors' research in an entire Discussion page. In addition, in the second paragraph, which describes the researches of other authors, the reference [26] are not in the literature list at all.

Answer: 26 is added in the reference now.

The results of the pot experiment, to which the largest and most difficult to understand Results section is devoted, have not been discussed at all. Such a single, meaningless sentence cannot be taken seriously: “On some of the studied physiological and root parameters of 21 d maize seedlings in controlled conditions had the influence both type of condition and type of fertilizer treatment.”\

Answer: The results of pot experiments description is now corrected and some sentences were changed.

Line 285. “According to other research, the use of increasing doses of the biostimulant provide …..” there are no references for other research.

Line 312-314. The sentence is difficult to understand, mainly because of the strange expression.

I would suggest avoiding saying “The Authors research”. Instead - our research, the research results, our experimental results etc.

In general, I would recommend making the Discussion part more focused. More purposefulness in data analysis is needed, linking them with the results of other authors' research. It should be clearly indicated what, according to the authors, would be ineffective, what would be promising, what could be used in practice.

The article is devoted to such a topical topic as the use of plant biostimulants of natural origin to promote plant growth and improve tolerance to a wide range of biotic and abiotic stress factors and is generally interesting. However, there are many inaccuracies in the article. The description of the methodology for the experimental variants and products used can definitely be improved. The Results section is too long and more attention should be paid to the correct statistical analysis of the results. The Discussion part should be more focused.  More purposefulness in data analysis is needed, linking them with the results of other authors' research.  I recommend accepting this article in Plants after major revision. I recommend that efforts be made to improve the scientific quality of this work, as the data material obtained in this study could be of interest to both plant physiology and agricultural scientists.

Answer: Methodology, statstical analysis and results section is improved now

Round 2

Reviewer 2 Report

The authors have improved the manuscript substantially. I am satisfied with the response of authors to my questions.

Reviewer 3 Report

Dear Author, 
The comments are addressed and paper can be subjected to acceptance.
Regards

Reviewer 4 Report

The authors have significantly improved the quality of the article and it can be published in Plants in present form.